# CNSL, a Promising Building Blocks for Sustainable Molecular Design of Surfactants: A Critical Review

**DOI:** 10.3390/molecules27041443

**Published:** 2022-02-21

**Authors:** Audrey Roy, Pauline Fajardie, Bénédicte Lepoittevin, Jérôme Baudoux, Vincent Lapinte, Sylvain Caillol, Benoit Briou

**Affiliations:** 1Orpia Innovation, CNRS, Bâtiment Chimie Balard, 1919 Route de Mendes, 34000 Montpellier, France; a.roy@orpiainnovation.com; 2Institut Charles Gerhardt Montpellier (ICGM), Université de Montpellier, CNRS, ENSCM, 34095 Montpellier, France; pauline.fajardie@univ-lyon1.fr (P.F.); vincent.lapinte@umontpellier.fr (V.L.); sylvain.caillol@enscm.fr (S.C.); 3Laboratoire de Chimie Moléculaire et Thio-Organique (LCMT), Normandie Université, ENSICAEN, UNICAEN, UMR CNRS 6507, 6 Boulevard Maréchal Juin, 14050 Caen, France; benedicte.lepoittevin@ensicaen.fr (B.L.); jerome.baudoux@ensicaen.fr (J.B.)

**Keywords:** CNSL, cardanol, surfactant, biobased

## Abstract

Surfactants are crystallizing a certain focus for consumer interest, and their market is still expected to grow by 4 to 5% each year. Most of the time these surfactants are of petroleum origin and are not often biodegradable. Cashew Nut Shell Liquid (CNSL) is a promising non-edible renewable resource, directly extracted from the shell of the cashew nut. The interesting structure of CNSL and its components (cardanol, anacardic acid and cardol) lead to the synthesis of biobased surfactants. Indeed, non-ionic, anionic, cationic and zwitterionic surfactants based on CNSL have been reported in the literature. Even now, CNSL is absent or barely mentioned in specialized review or chapters talking about synthetic biobased surfactants. Thus, this review focuses on CNSL as a building block for the synthesis of surfactants. In the first part, it describes and criticizes the synthesis of molecules and in the second part, it compares the efficiency and the properties (CMC, surface tension, kraft temperature, biodegradability) of the obtained products with each other and with commercial ones.

## 1. Introduction

Our world and the societies that we have built are going through a major phase of great change punctuated by increasingly frequent and violent crises. The threat of climate change and all the economic, political, and ecological upheavals it brings, impact collective consciousness, and influence future development strategies. Not only consumers, but also countries and industries, are increasingly sensitive to the origin of the products they buy, the impact of their production processes, their effects on human health and on the environment. They are also concerned about the end of life of these products, their possible recycling, or the pollution they can cause. The world of chemistry, the nerve center of the creation of our daily products, is naturally alert to these questions. The use of and access to fossil resources, increasingly complex or even controversial, are gradually giving way to the biobased building blocks [1,2].

Numerous studies are bringing to light the possible toxicity of currently widely used products such as pesticide [3], and additives [4,5,6]. This has the effect of accelerating the search for alternatives that are as safe as possible for human beings and the planet on which we live. It is in this momentum that a sustainable and green chemistry is emerging and defined [7,8]. The aim and the definition of sustainability are to reconcile economic, environmental, and social dimensions with the aid of the 3Ps: People, Planet and Profit, around the same objective, and to ensure its continuity over time [9,10]. 

In these times, when components and additives are deeply studied, scrutinized, and criticized, surfactants are crystallizing a certain focus for consumer interest. Present in our shampoos, our detergents, our household products, our cosmetics, and also in the world of construction and materials, their omnipresence and their possible impact are worrying. Their proximity to our body, our food, and the water we consume make surfactants a potential source of pollution and danger for humans, but also for rivers, oceans, sealife, wildlife, and flora on which we depend, and for which we are responsible [11,12,13]. 

For the past ten years, we have seen the blooming of possible alternatives to current industrial surfactant products. There is also an upsurge in homemade soap, laundry, cosmetics, and detergents. The compositions of which are made up of easily accessible and natural ingredients. In contrast, within academic research (Figure 1), there is an increase in articles about what are called “green surfactants”.

Many bioresources such as sugar [14,15], vegetable oil [16,17] or lignin [18] have been already used to synthesize and produce efficient surfactant. Some of them such as Alkyl PolyGlycoside (APG), or glycerol esters are already commercially available. However, vegetable oil and sugar are feedstock, and compete with animal and human alimentation. Moreover, the actual production of beetroot for sugar induces the use of neonicotinoid, poison for bees and other important insect pollinizers [19,20]. In addition, a so-called green or sustainable surfactant is, of course, a biobased molecule. However, the term “green” is not limited to the source of this raw material. It is good to remember that “bio-based” or “natural” does not equal “non-hazardous” or “non-toxic”. “Green” also relates to the manufacturing process, the low degree of dangers or toxicity of both reagents and final product, and the end of life of the product [8]. 

This topic is complex. Moreover, some so-called “biobased” products are often only partially biobased. Finally, the “green” solutions proposed must be able to be industrialized in the long term without endangering the wildlife and without competing with crops linked to human or animal food.

CNSL (Cashew NutShell Liquid) is a versatile abundant bioresource already used to synthesize surfactants. Even now, CNSL is absent or barely mentioned in specialized review or chapters talking about synthetic biobased surfactant [21,22,23]. A lot of work on this raw material has been undertaken in this field since the last, and only brief, review focused on CNSL [24]. Given the growing interest in this resource over the past decade, it seems relevant to summarize the work carried out. The aim, throughout this review, is to show, detail and criticize the work already made with this promising resource for the synthesis of biobased and green surfactants, and project perspectives for the future of this raw material in this field.

After a general introduction about what, exactly, CNSL is, we will first describe and criticize the synthesis of surfactants currently made using this feedstock. Next, we will discuss the properties of these surfactants and then, finally, we will give our conclusions and our perspectives on the future advances of these green CSNL-based surfactants.

## 2. Cashew NutShell Liquid

Cashew NutShell Liquid (CNSL) is an oil extracted from a by-product of cashew cultivation, currently considered as waste: its shell. Inedible, and not competing with food production, the tonnages of the nutshells are closely linked to that of the edible cashew kernels already produced. The annual world production of cashew nut shells is estimated at around 3M T/year and is located in geographical areas with a tropical climate such as South America (Brazil), Asia (India, Vietnam, etc.) and Africa (Côte d’Ivoire, Benin, Burkina Faso, Tanzania, etc.) [25,26]. Various extraction methods make it possible to obtain CNSL (pyrolysis [27], solvent extraction [28], screw pressing [29] and supercritical CO_2_ extraction [30,31]) with mass yields oscillating between 15 and 30%. When we talk about CNSL, we must distinguish two different types of oils by their composition: the CNSL says “natural” and the one called “technical” (Figure 2).

Natural CNSL is obtained under gentle or low temperature extraction conditions. The oil obtained is mainly composed of anacardic acid (60–70%), cardol (10–20%) and cardanol (<10%). Anacardic acid tends to decarboxylate into cardanol around 140 °C [32]. This decarboxylation can take place during extraction under heating or pressing conditions. It yields a technical CNSL, mainly composed of cardanol (70–80%) and cardol (10–20%). In both CNSLs, there are also methyl cardol (<3%) and traces of urushiol. Some extraction processes also tend to promote the formation of residol, a polymeric residue from the polycondensation of CNSL phenolic compounds. It is possible to separate and purify the various compounds of CNSL by precipitation [33], distillation [34], liquid/liquid extraction [33,35] or chromatographic column [36].

The three main compounds of CNSL, anacardic acid, cardol and cardanol can be each found in four forms: saturated, monoene, diene and triene. Unsaturated or mixed compounds are present in oil form, whereas hydrogenated compounds are in the solid form.

Due to their chemical structure, CNSL compounds are biobased synthons of choice for chemical synthesis (Figure 1). First, they are natural and non-toxic phenolic compounds allowing Mannich reactions [37,38,39], polycondensation [40,41], benzoxazine formation [42,43] or any other reaction occurring onto an aromatic ring. The hydroxy function of phenol allows many functionalization reactions such as esterification [44,45], etherification [46,47], or nucleophilic substitution [48,49]. Finally, the hydrophobic alkyl chain in C_15_, when it is unsaturated, in addition to providing a plasticizing effect [41,42,50], also allows many chemical reactions: epoxidation [44,46,47], hydrogenation [51,52], carbonation [53], thiolene [54,55] or Diels Alder [56].

For all these reasons, we find the compounds of CNSL, and more particularly cardanol, easier to obtain in large quantities, used in the synthesis of additives (plasticizer [39,44,47,50,57], lubricant [58,59]), polymers [53] (epoxy [46,60,61], phenolic resin [40,41,62], polyurethane [55,63] etc.) or active molecules [64]. Anacardic acid, with its structure close to salicylic acid (Figure 3a) and cardol, close to the structure of resorcinol (Figure 3b) have very relevant intrinsic biological activities [65,66,67,68,69,70,71]. 

Due to its versatility, abundance, easy functionalization, and biological activity, CNSL and the molecules that compose it are building blocks of choice for the development of surfactants, whether for application in emulsion synthesis, for detergent or as a formulating agent.

## 3. Synthesis of CNSL-Based Surfactants

### 3.1. Introduction

Surfactants are amphiphilic molecules widely used in industry. The global surfactant market is estimated at $41.3 billion in 2019 [72]. Their peculiar structure, composed of both lipophilic and hydrophilic parts in the form of nano-domains (micelles, etc.), to adsorb at interfaces where they reduce interfacial or surface tension. This property makes them very interesting for a wide range of applications such as household detergents, personal care, industrial and institutional cleaners, food and beverage, etc., [73].

Surfactants are often used in combination with water and can easily be released into the environment, disrupting the ecosystem. In the current context intimately linked to public health and environmental protection issues, the development of bio-based and non-toxic surfactants seems inevitable and relevant. In recent years, the surfactant market has been evolving due to an increasing consumer demand for bio-based products. Thus, surfactants of natural origin represent an important marketing issue for the development of different industrial formulations.

There are two categories of surfactants: nonionic and ionic. The latter are themselves divided into three categories: anionic, cationic and zwitterionic. 

### 3.2. Anionic Surfactants Based on CNSL

The CNSL-based anionic surfactants are divided into sulfonate and carboxylate compounds and are mainly based on cardanol with rare examples on anacardic acid derivatives.

#### 3.2.1. Carboxylated CNSL-Based Surfactants

Two routes have been investigated to synthesize this class of surfactants, either starting from a natural phenol bearing a carboxylic acid (anacardic acid) or converting the natural phenol group into carboxylic acid by carboxymethylation.

The simplest carboxylated surfactants were the anarcardate salts resulting from the deprotonation of acid group by KOH (98% yield) (Figure 2). They were compared to 2-o-cardanyl acetate sodium salt, produced in a low yield (41%) by carboxymethylation using Williamson reaction between the phenol unit of cardanol and sodium chloroacetate [74,75]. Compared to the first natural one, this second surfactant was 85% biobased, and uses an irritant and environmentally toxic reactant. More recently benzoxazine surfactants were synthesized via a Mannich reaction using amino acids (glycine) and cardanol [76]. 

Some carboxylated CNSL-based surfactants have a pendant polymeric chain as spacer between the ionic unit and the aromatic ring (Figure 3). A first category of surfactants was based on poly(ethylene oxide) (PEO) starting from saturated or unsaturated cardanol [77,78]. Unsaturated cardanol PEO ether carboxylated (CPEC) and its saturated counterpart (SCPEC) have been synthesized in two steps. The first step consisted in the reaction with ethylene oxide, a CMR extremely inflammable and explosive gas, on phenol to produce the cardanol PEO ethers (CPE), reducing the biobased nature of this surfactant to 40–30%. The second step consisted in the functionalization of the PEO terminal alcohol group into carboxylate group using chloroacetic acid. Polymeric cardanol surfactants were extended to pluronic combining hydrophilic PEO block polymer to hydrophobic poly(propylene oxide) (PPO) block (PO:EO 1:15 to 25) [79]. 

Using the strategy of CPEC, Gemini saturated cardanol surfactants (GSC) were produced in 95% yield after adding a supplementary step of dimerization of cardanol using formaldehyde, a CMR agent, in the presence of HCl [80]. Various salts (Na, K, Ca, Mg) were described [81].

#### 3.2.2. Sulfonated CNSL-Based Surfactants

The most abundant anionic cardanol surfactants are the sulfonated ones with the polar head linked to the aromatic or at the end of an aliphatic chain (with a molecular or polymeric spacer between the cycle and the ionic head).

##### Aromatic Sulfonated Surfactants

The most direct route to reach sulfonated cardanol surfactant was the sulfonation of the aromatic ring to afford a mixture of two isomers in ortho and para position in 86% yield with 80% biobased. (Figure 4) [82,83,84,85]. It was compared to sodium n-alkylbenzenesulfonate as dodecylbenzene sulfonate (DDBS) [86] and diversified into calcium [87] and potassium [88] salts. The phenol ring was also converted into acrylate group to elaborate saturated cardanol surfmers using acrylate chloride (87% yield) followed by sulfonation using chlorosulfonic acid (65% yield) [89]. An alternative was also described with the addition of a spacer between the phenol ring and the ionic group using 2-chloroethanol, reducing the biobased content under 70% and using more toxic reactant for no real improvement. 

Some patents described 60% biobased monosulfonated cardanol derivatives containing siloxane unit (Figure 5). A first example resulted from the reaction between sulfonated cardanol and (3-chloropropyl)trimethoxysilane [90]. Similar structures were elaborated with epoxidized cardanol and a trimethoxysilane primary amine coming from the monoaddition of diamine (n = 2–8) on (3-chloropropyl)trimethoxysilane [91,92]. In that case, epichlorohydrin, a toxic and CMR substance, was added on cardanol to introduce epoxide unit prior to the reaction with the amine. Eventually, the surfactant obtained was 40% to 60% biobased.

##### Gemini Aromatic Sulfonated Surfactants

Ester-linked anionic Gemini surfactants were investigated using various lengths of diacid chain (from 4 to 6 carbons) by esterification of phenol followed by a regioselective sulfonation in the ortho position (Figure 6) [83]. Otherwise, ether-linked anionic Gemini surfactants were studied after sulfonation of cardanol in presence of highly reactive oleum followed by etherification with various toxic dichloroalkanes [93]. Another pathway was the reaction of epoxidized cardanol with various diamines producing a surfactant that was 50% biobased [94].

##### Aliphatic Sulfonated Surfactants

In contrast to the above-mentioned surfactants, the present anionic molecules contain sulfonate group in alkyl chain of the spacer instead of aromatic position (Figure 7). Cardanol was directly transformed by highly toxic 1,3-propanesultone [95], or after the reaction with 3-chloropropylamine hydrochloride (PPA) for a second time, gave PSA-PPA [48].

The strategy of sultone was employed to modify cardanol poly(oxyethylene) ethers (CPE) (Figure 8), where an aliphatic alkoxide opened 1,3-propanesultone or 1,4-butanesultone to generate sulfonate end group [96] with different lengths of PEO. The use of sultone (1,3-propanesultone or 1,4-butane sultone), nevertheless, implies the addition of CMR molecule to the synthesis of these surfactants. Another methodology of sulfonation was employed with the ring opening of maleic acid by the activated terminal alcohol of PEO, followed by the sulfonation in presence of sulfurous acid sodium to yield cardanol PEO ether sulfosuccinic acid monoester disodium salt [97]. No data have been reported on the behavior in water.

To conclude, for cardanol sulfonated surfactants, sulfonate group is either directly on the aromatic ring or at the end of an alkyl or polymer chain. Gemini sulfonate surfactants were also reported. However, some of the reactants used (epichlorohydrin acryloyl chloride (lacrimal product), sodium chloroacetate, sultones,) used are toxic or CMR, hence some of the synthesis routes should be improved. In addition, what is often called bio-based surfactant is in fact, very often only partially (or even only half) bio-based.

### 3.3. Cationic Surfactants Based on CNSL

This section summarizes the different pathways to functionalize cardanol by a cationic group to obtain amphiphilic molecules. The first data on cationic derivatives appeared in 1964; this preliminary research was followed by several publications and patents. The cationic part of these molecules was exclusively represented by ammonium, pyridinium and imidazolium groups, which we will describe in this section.

#### 3.3.1. Ammonium Derivatives of Cardanol

Cashew nutshell liquid (CNSL) has been used for the first time to prepare quaternary ammonium salts with the aim of obtaining water soluble compounds with germicidal properties [98]. 

Several decades later, quaternary ammonium salts used as a phase transfer agent were prepared in five steps starting from cardanol (Figure 9).

Methylated cardanol was obtained by reaction with dimethylsulfate (CMR) in basic conditions, then, ozonolysis afforded 3-methoxyphenyl-1-octanaldehyde in 50% yield. Treatment with hydroxylamine hydrochloride gave the oxime, and the corresponding amine hydrochloride was obtained by hydrogenation in halogenated solvent (CHCl_3_). The primary amine hydrochloride was treated with different iodide compounds (methyl, ethyl and propyl) to give the quaternary ammonium salts (C-1a, C-1b and C-1c) [99]. This synthesis involved too many steps, using a lot of hazardous products (ozone, dimethyl sulfate, hydroxylamine) to obtain a half biobased surfactant. In view of its structure, the individual toxicity of this surfactant should also be studied. 

More recently, in the field of composites, an 66% biobased ammonium surfactant has been used to modify the surface of sepiolite clay and to enhance its compatibility with epoxy networks [100]. It was synthesized from commercially available epoxy derivative of cardanol in two steps (Figure 10). First, the neutral form of surfactant was obtained by ring-opening of the epoxide with 1,2-diaminoethane. Then, the cationic form was obtained by protonation with chlorohydric acid.

In 2016, symmetrical Gemini surfactants and unsymmetrical surfactants were prepared from cardanol [101]. In this research, ammonium cardanol-based surfactants (eventually including a benzyl bromide group) were synthesized in three steps. These surfactants were obtained by etherification with 1,4-dibromobutane followed by nucleophilic substitution with dimethylamine. Next, the tertiary amine reacted with dibromo compounds (as 4,4′-di(bromomethyl)benzophenone) to form the (di)cationic surfactants. According to the reaction stoichiometry, a monocationic (39% biobased) or Gemini surfactant (51% biobased) could be obtained. The Gemini surfactant was used as emulsifier in radical emulsion polymerization of methyl methacrylate. The monocationic surfactant with benzyl bromide group acted as both emulsifier and initiator for AGET (activator generated by electron transfer) ATRP (atom transfer radical polymerization) emulsion polymerization (Figure 11).

In the same study, others 50–60% biobased Gemini cationic structures were reported and used as stabilizers for the radical emulsion polymerization of methyl methacrylate (Figure 4).

Using a similar procedure, two 60% biobased cationic surfactants were prepared by Luo et al. [102] by etherification reaction with 1,2-dibromoethane (CMR) or 1,6-dibromohexane. Then, cationic group was introduced by nucleophilic substitution with trimethylamine (Figure 12).

With the same aim to develop antimicrobial applications of hydrogenated cardanol-derived quaternary ammonium compounds, another study reported the preparation of seven structures, including heterocyclic or non-symmetrical Gemini salts with double bond or carbon chain with hydroxyl group [103].

A cationic surfactant was prepared in two steps from hydrogenated cardanol. An alkylation of phenol was performed by reaction with 1,4-dibromobut-2-ene, a toxic reagent, in basic conditions with a reaction yield equal to 54%. Thus, reaction with the heterocycle N-methylmorpholine gave the corresponding cationic surfactant with a yield of 89.4% and a biobased origin of 52% (Figure 13).

Non-symmetrical Gemini ammonium surfactants were prepared by reaction of the brominated derivative with butyl or hexyl salts obtained from 1,4-diazabicyclo[2.2.2]octane (Figure 14). The reaction yields were equal to 72% (butyl) and 76% (hexyl) and led to around 40% biobased compound.

Finally, using a different method, the same authors reported the preparation of half biobased cationic surfactants with hydroxyl functions and heterocycles. The alkylation of hydrogenated cardanol was performed with epichlorhydrin, a CMR reactant, followed by epoxy ring opening with protonated N-methylmorpholine (Figure 15).

In 2020, using hydrogenated cardanol (3-pentadecyl phenol), similar structures (Figure 5) were obtained by reaction with 1,2-dibromobutane (CMR), followed by quaternization reaction with different tertiary amines (trimethylamine, triethylamine, N-methyl morpholine and N-methyl piperidine) [104]. Some of them are quite toxic or harmful.

Theses 60–65% biobased cationic surfactants were assessed for their biocompatibility and their potential to solubilize curcumin as a model drug, which has poorly water solubility.

A polyfunctional half biobased cationic surfactant was synthesized from cardanol in very good yield after 2 steps. This sequence began by simple alkylation of the phenol by a halogenated aliphatic amine under basic conditions. The primary amine of the cardanol derivative was used for an opening reaction of N-trimethylammoniumglycidyl chloride, a CMR molecule, in 96% yield. Thus, an ammonium was introduced in this step to generate a new surfactant bearing two other polar functions with a secondary amine and an alcohol (Figure 16) [48].

The polyfunctionality of this cationic surfactant is a real advantage with a very interesting solubility in pure water.

Very recently, a patent followed by a publication provided a series of 45–65% biobased hydroxylated cardanol ammonium salts with one or two hydroxyl groups [105]. These surfactants were designed and investigated as additive in detergent [106]. Hydroxylated cardanol ammonium salts were obtained by reaction of cardanol with epichlorhydrin in basic conditions. Then, after preparation of epoxide derivate with 59% reaction yield, cationic hydroxylated structures were obtained by reaction with different tertiary amines. Various dangerous reactants were involved in the synthesis of these products such as epichlorohydrin, dimethylbutylamine, TMEDA and dimethylaminoethanol. The chemical scheme and the six cationic structures are presented in Figure 17.

Cardanol was modified to N-benzyl-N,N-dimethyl-3-pentadecylclohexan-1-ammonium, in 2 steps with an overall yield of 51% (Figure 18). The use of hydrogen under pressure in presence of Pd/C and dimethylamine led to a reductive amination reaction in 85% yield. The previous aliphatic amine is a very interesting starting material of various cationic surfactants but not only (see section on zwitterions). Indeed, a simple alkylation by benzyl chloride, unfortunately toxic, of the tertiary amine afforded ammonium in 62% yield as 68% biobased new cationic surfactant [107].

Another patent reported the preparation of 60% biobased cardanol-derived ammonium surfactants by epoxy ring opening reaction with different (sometimes toxic) allylamine as diallylmethylamine, triallylamine or diallylamine, the chemical structures are represented below (Figure 6) [108]. These surfactants were copolymerized in aqueous solution with methacrylic monomers and redox initiator system.

#### 3.3.2. Macromolecular Ammonium CNSL-Based Surfactants

Amphiphilic poly(ionic liquid)s C-29a and C-29-b (Figure 19) were obtained from an acrylamide monomer bearing acid sulfonic group, the 2-acrylamido-2-methyl-1-propane-sulfonic acid [109]. In the two first steps, the phenol group of cardanol was etherified with diethanolamine, ethanolamine and tetraethylene glycol using a linking agent based on bis(2-chloroethyl)ether. Then, the amine group was quaternized by acid base reaction with the monomer and the radical polymerization was performed with 2,2′-azobis(2-methylpropionitrile) (AIBN) as initiator at 70 °C (C-29a). The surface activity of the polymer was studied with the aim to use it as demulsifier for crude oil water emulsions. This study reported the preparation of another amphiphilic poly(ionic liquid) bearing three hydrophilic chains (molecule C-29b not described in this section).

In a recent research work, an antimicrobial cationic agent obtained in three steps from cardanol was embedded in UV curable epoxy acrylate system to develop antibacterial coatings [110]. The bromination was carried out by reaction of bromine to double bonds of cardanol, then, quaternary ammonium salt was obtained by reaction with triethylamine. In the last step, ring opening reaction with glycidyl methacrylate (CMR) gave the corresponding dicationic methacrylic monomer able to copolymerize with epoxy acrylate (Figure 20). At the end, this monomer was only 25% biobased, brough by the cardanol part.

#### 3.3.3. Pyridinium Derivatives of Cardanol

Two studies report a similar procedure for obtaining 62% biobased cationic surfactants by introducing pyridinium group to cardanol in two steps [102,104]. Firstly, cardanol was used for the nucleophilic substitution of an alkyl halide in the presence of base (K_2_CO_3_ or KOH). Further, the pyridinium salts were obtained after reaction with pyridine, not the friendliest solvents (Figure 21).

Hydroxylated cardanol pyridinium salt was obtained in 59% yield by reaction of cardanol with epichlorhydrin in basic conditions (Figure 22). Then, the epoxide afforded a 63% biobased cationic hydroxylated surfactant by reaction with pyridine hydrochloride in 88% yield [106]. Again, epichlorohydrin and pyridine were involved in the synthesis despite their dangerousness for human health.

A series of half biobased Gemini pyridinium amphiphiles were synthesized by a two-step procedure starting from 1-(allyloxy)-3-pentadecylbenzene (Figure 23) [111]. In the first step, a bifunctional cardanol was obtained by regioselective bromination of allylated cardanol with different dithiol spacers. Then, amphiphilic products were prepared by quaternization of pyridine.

Recently, a patent described the preparation of different surfactants with bipyridinium polar groups. After alkylation of hydrogenated phenol with 1,4-dibromobut-2-ene, the surfactants were obtained by a second alkylation reaction with different salts obtained from 4-4′-bipyridine (Figure 24) [109]. In the end, these pyridinium salts contained a slightly biobased part, only provided, again, by cardanol.

#### 3.3.4. Imidazolium Derivatives of Cardanol

Using the same procedure as previously described [111], a series of half biobased gemini imidazolium amphiphiles were synthesized by a two-step procedure starting from 1-(allyloxy)-3-pentadecylbenzene. (Figure 25.) In the first step, a bifunctional cardanol was obtained by regioselective bromination of allylated cardanol with different dithiol spacers. Then, amphiphilic products were prepared by quaternization of *N*-methylimidazole.

A second study reported the preparation of imidazolium-based ionic liquid from cardanol [112]. After distillation and reduction of double bonds of alkyl chain, a brominated spacer was introduced by reaction with 1,4-dibromobutane under basic conditions. Then, quaternization reaction of 1-methylimidazole was realized and the bromide salt was converted into hexafluorophophosphate salt by anion metathesis reaction (Figure 26).

The amphiphilic character of this product was not directly mentioned by the authors, but they used its self-assembly properties to design thermotropic and lyotropic phases. This 50% biobased ionic liquid crystal is an efficient electrolyte for energy storage devices.

Several cationic surfactants were, essentially, synthesized from cardanol, which was almost the only molecule bio-based in these reactions. In addition to the use of CMR or toxic reagents, the structure of these molecules such as quaternary ammonium leads us to reflect on the possible toxicity of these new surfactants. Unfortunately, little information on this subject, to our knowledge, has yet been disclosed.

#### 3.3.5. Pyrazolium and Thiazolium CNSL-Based Surfactants

Pyrazolium and thiazolium surfactants were prepared in two steps from hydrogenated cardanol. As previously described, alkylation of phenol by 1,4-dibromobut-2-ene in basic conditions gave halogenated cardanol derivative (Figure 13). Then, reaction with 1-methylpyrazole or thiazole gave the corresponding cationic surfactants with reaction yield equal to 86.3%, and 79.3%, and a biobased origin of 58%, and 57.5% respectively (Figure 27).

A second example of thiazolium surfactant was obtained in two steps by reaction of cardanol with epichlorohydrin followed by epoxy ring opening with protonated thiazole (Figure 28).

### 3.4. Zwitterionic Surfactants Based on CNSL

Zwitterionic surfactants are organic compounds exhibiting an apolar aliphatic chain and a polar head group with a positive and negative charge. This particularity provides them very advantageous properties compared to conventional cationic or anionic surfactants. Usually, zwitterionic surfactants possess valuable antibacterial and fungicidal activity combined to good compatibility with a large range of surfactants. These salts are known for their robust structure, high tolerance to salinity and temperature. Moreover, zwitterionic surfactants have low toxicity which makes them almost non-irritating to the skin and eyes. Thus, these foaming agents can be used in mild shampoos and skin cleansers. Several other interesting and complementary properties should also be noted: these zwitterions are classified as antistatic and emulsifying agents with a good dispersibility, wettability and biocompatibility.

To date, very few examples have described zwitterionic surfactants derived from CNSL oil. All studies used cardanol as a raw material with a cationic part resulting from a quaternized amine or introduced by a ring opening. Concerning the anionic part, a carboxylic or sulfonate anion was mainly used. To this end, phenol represents a very promising reactive function for developing zwitterionic surfactants by simple etherification. From this postulate, some pathways were developed in recent years, which we will summarize in this section.

Several 3-Pentadecylcyclohexylamines were synthesized from CNSL by reductive amination in water [107]. In this ecologically benign polar and protic solvent, the use of Pd/C catalyst at 5–10 bars hydrogen pressure gave aliphatic amines in moderate to good yield after 15 h at 100 °C. The presence of dimethylamine or dibutylamine, which is more toxic than the other one, (1.1 equiv), a 20% higher yield was similarly observed (Figure 29). In these two cases, the scalability of the reaction was studied successfully over several grams (15–20 g). The oxidation of previous amines by using hydrogen peroxide (3 equiv) gave N-oxide 83% biobased zwitterionic surfactants Z-1 in excellent yield without purification. Finally, dimethylated amine was also quaternized by addition of toxic bromoacetic acid (1 equiv) in ethanol-H_2_O. In presence of NaHCO_3_, betaine 2-(dimethyl(3-pentadecyl-cyclohexyl)ammonium)acetate Z-2 was prepared in moderate yield (light brown solid). This synthesis led to a 74% biobased aliphatic betaine in only two steps.

Recently, zwitterionic surfactant based on cardanol was synthesized in three steps with an overall yield of 65% (Figure 30) [48]. First, nucleophilic substitution of 3-chloropropylamime in excess of NaOH gave a primary amine in 94% yield. At reflux of methanol, a ring-opening reaction of toxic propanesultone (1 equiv) was carried out to give N-propanesulfonic acid in 75% yield. Then, the secondary amine of the previous compound was used to generate polyfunctional compound Z-3 in 92% yield from commercially available *N*-trimethylammoniumglycidyl chloride (1 equiv). This sequence afforded a poorly (46%) biobased surfactant as a mixture with predominantly an unsaturated aliphatic chain of one to three alkenes.

In 2019, a series of copolymers containing sulfobetaine and cardanol was synthesized via free radical polymerization of methacrylate derivatives (Figure 31) [113]. After ring-opening reaction of glycidyl methacrylate by cardanol, the aromatic compound having methacrylate moiety was added to a solution of (dimethylamino)ethylmethacrylate and AIBN in different ratios. After 18 h at 65 °C, all the resulting solutions were cooled down and poured into water/methanol to precipitate the synthesized copolymer in moderate to good yield (>50%). After quaternization of some tertiary amines by propanesultone, Z-4 type zwitterionic compound was prepared in THF at room temperature The disadvantage of this synthesis is the use of three different CMR compounds.

In 2015 [114], zwitterionic surfactants were described from saturated cardanol without chemical modification of the phenol. In this research, a regioselective Mannich reaction of hydrogenated cardanol with formaldehyde (37% in water) and dialkylamine R_2_N (R = methyl, ethyl, propyl or butyl) was achieved giving cardanol dialkylamine in very good yield. Figure 32 describes the synthesis of two different surfactants, where dialkylamine of cardanol was substituted by a methyl or ethyl residue. A 72% biobased saturated cardanol betaine surfactant Z-5 was synthesized by nucleophilic substitution of sodium chloroacetate in alcoholic medium and water to obtain a yellow-brown solid. Furthermore, saturated diethyl ammonium sulfonate betaine surfactant Z-6 was prepared by quaternarization with 3-chloro-2-hydroxypropane sulfonate leading to a 58% biobased surfactant. In order to complete the synthesis of surfactants with a good yield, the choice of solvent (short-chain alcohol) and the concentration of cardanol (0.1 to 5 mol/L) are essential.

Very recently, two patents provided a temperature-resistant salt-resistant cardanol zwitterionic surfactants [115,116]. In the preparation method, the first step required an excess of toxic epichlorhydrin in presence of a phase transfer catalyst such as Bu_4_N^+^Br^−^, Bu_4_N^+^Cl^−^, BnEt_3_N^+^Cl^−^, C_12_H_25_Me_3_N^+^Cl^−^ or C_14_H_29_Me_3_N^+^Cl. Depending on the catalyst, cardanol was heated in alcoholic solvent (Methanol, Ethanol, Isopropanol or Isobutanol) to give cardanol chlorohydrin ether in very good yield. To the previous compound were added diphenylamine (excess) and organic base (Et_3_N or pyridine) in similar protic solvent to afford tertiary amine after heating for a few hours. The last step used sodium 3-chloro-2-hydroxypropane sulfonate (excess) as alkylating reagent to yield a 66% biobased zwitterionic surfactant. The Figure 33 describes a sequence leading to compound Z-7 with an overall yield of 64%.

Based on a very similar strategy, another sequence was described to provide a 74% biobased cardanol surfactants in three steps (Figure 34). To this end, tertiary amine compounds were synthesized from various disubstituted amine (CH_3_- to C_5_H_11_-) and cardanol. As an example, tertiary amine (1 equiv) and 4-chloro-3-hydroxy-butyric acid (2 equiv) were heated in ethanol several hours to give zwitterionic cardanol Z-8 in good yield.

To date, our knowledge about the zwitterionic surfactant is limited to these examples with few information on their properties and applications. All the syntheses require two or three steps to generate the polar head. We can note that all these salts have an alcohol and/or phenol function, in addition to the positive and negative charges. To facilitate the synthesis and purification of the products, these different studies employ the same strategy by using the final step to simultaneously generate the two charges on the cardanol derivative. All these syntheses involve the use of common, but toxic, and/or CMR, reagents such as epichlorohydrin, 1,3-propanesultone, glycidyl methacrylate and so on. These sequences should be revisited to be more ecologically friendly. In addition, the cardanol block-building is almost the only biobased part of the surfactant that sometimes halves the biobased contribution in the synthesis of the surfactant.

### 3.5. Nonionic Surfactants Based on CNSL

Nonionic surfactants are molecules that do not undergo ionization when being dissolved in water. They exhibit high stability in solution, good solubility in both water and organic solvents, and excellent compatibility with other surfactants. This section describes the different synthesis of nonionic surfactants based on CNSL. All these surfactants have a hydrophilic part such as poly(ethylene oxide) or PEO, polyoxazoline and/or sugar groups. According to the nature of the polar head, nonionic surfactants can be partially or totally biobased.

#### 3.5.1. Partially Biobased CNSL-Based Surfactants

These molecules are made of cardanol or cardol as the biobased hydrophobic part and PEO or polyoxazoline as the non-biobased polar head.

##### CNSL-Based Surfactant with Ethoxy Head

Ethoxylation of cardanol [117,118,119,120,121,122,123,124,125] or cardol [117,122] is the main route to synthesize this kind of surfactant (Figure 35).

The reaction required opening of ethylene oxide by cardanol/cardol between 120 and 220 °C, mostly in base-catalyzed conditions [117,118,119,121,123,124]. NaOH and KOH are mainly used as catalyst, but the acidity of these phenols allowed the use of K_2_CO_3_, Ca(OH)_2_ or NaHCO_3_. Ethoxylation was also carried out with acetic acid as a catalyst. The degree of ethoxylation (m), and thereby the length of the hydrophilic part, was controlled by the molar ratio between cardanol and ethylene oxide [120] and/or the duration of the reaction [117]. For example, Tyman et al. [117] synthesized polyethoxylates from cardanol (NI-1) and cardol (NI-2), with a saturated or unsaturated aliphatic side chain, and with a degree of ethoxylation m = 1–48. The biobased part of these surfactants decreased from 87% to 13% when the length of PEO chain increased. The surface behavior of such molecules depended on the number of ethoxy units and the nature of the phenol. Properties and/or applications of ethoxylated CNSL-based surfactants are further described in the second part of this review.

Nonionic Gemini surfactant based on cardanol was synthesized in a two-step reaction [126] (Figure 36).

The first step consisted of a Mannich reaction between cardanol, formaldehyde and methylamine as toxic and corrosive reagents to form a cardanol dimer. The second step was an ethoxylation of the cardanol dimer in basic conditions. Nonionic Gemini surfactants (NI-3), with m + p = 6–60, were thus obtained, with different cloud points according to the length of the polar head. The biobased part of such molecules can be decreased with the length of the polar head, from 65% (m + p = 6) to 18% (m + p = 60).

Branched ethoxylated surfactants based on cardanol can also be obtained through ethoxylation and reaction with glycerol. Here is one example of such compound, described in a Chinese patent [127] (Figure 7).

A sequential, one-pot synthesis was carried out to make this product. First, cardanol was propoxylated with very toxic propylene oxide (CMR) in basic conditions at 140 °C for 1 h. Then, glycerol was added to the medium, and the reaction was left at 100 °C for 1 h. Finally, the intermediate product is ethoxylated again by ethylene oxide at 140 °C for 5 h. The final product (28% biobased) was obtained after neutralization by acetic acid. Other structures can be obtained according to the nature of the reactants (ethylene, propylene or butylene oxide, glycerol or glycidol) and/or their molar ratio.

Ethoxylated cardanol surfactants can be further functionalized with siloxane derivatives [128] (Figure 37).

Ethoxylated saturated cardanol was reacted with chloropropyl trisiloxane with catalytic amount of NaOH at 85–105 °C during 6–8 h. The biobased part of the final products was comprised between 31% and 35%.

All these surfactants were synthesized with ethylene oxide (or a derivative such as propylene oxide), a very toxic precursor, mostly produced with petroleum-based ethylene. Some authors developed alternative strategies to synthesize ethoxylated cardanol surfactants without using ethylene oxide. For example, Rahobinirina et al. [129] developed an ethoxylated compound based on 3-pentadecylcyclohexanone, an intermediate obtained from CNSL (Figure 38).

The first step consisted of a hydrogenation of technical CNSL under pressure (20 to 30 bars) at 80 °C for 3 h in presence of Pd/C. In these conditions, a complete reduction was observed. 3-pentadecylcyclohexanone was thus obtained in 67% yield. The second step was a reductive alkylation of the intermediate with triethylene glycol (5 eq.) at 100 °C for 14–28 h in moderate yield. The final product was 70% biobased. One can notice the use of hexane, a CMR compound, as solvent. According to the authors, this ethoxylated derivative of CNSL could be used as surfactant. Still, its surface tension properties were not studied.

Nonionic CNSL-based surfactants can be obtained through the addition of a large excess of very toxic ethylene carbonate, followed by decarboxylation at high temperature [130] (Figure 39).

The reaction between hydrogenated cardanol and ethylene carbonate occurred at 190 °C for 5 h in the presence of PPh_3_. Ethylene carbonate conversion was estimated at 96.9% by ^1^H NMR. 27% biobased ethoxylated cardanol NI-7 was thus synthesized. Feitosa et al. [130] also prepared an ethoxylated cardanol formaldehyde resin (ECFR) in two steps. First, hydrogenated cardanol was reacted with toxic formaldehyde in acidic conditions at 130 °C for 4 h. Then, the same procedure as the one applied for the synthesis of ethoxylated cardanol was employed to obtain ethoxylated cardanol formaldehyde resin. Ethylene carbonate can be considered as less harmless compared to ethylene oxide; yet, it is industrially produced using this toxic reagent.

Atta et al. [131] prepared two cardanol ethoxylate amine-based derivatives, as depicted in the following Figure 40.

The di-etherified cardanoxy amine (DECA, NI-8) was obtained by etherification process in two steps. First, cardanol was etherified à 150 °C with β,β-dicholorodiethylether (DCDE), and diethanolamine in basic medium. Then, the hydroxy amine group of the product were etherified with DCDE and tetraethylene glycol (TEG) in the presence of NaOH at 100 °C for 5 h., whereby 30% biobased DECA was synthesized with an overall yield of 90.2%.

The tri-etherified cardanoxy amine (TECA, NI-9) was obtained in three steps. Cardanol was etherified with an excess of epichlorohydrin in basic conditions. The reaction was carried out in water at reflux for 9 h. Then, the epoxide was opened by 1 equivalent of ethanolamine at 100 °C. The last step consisted of the etherification of the hydroxy amine groups of the derivative with DCDE and TEG. The global procedure was similar to the one described for DECA. 25% biobased TECA was obtained with an overall yield of 93.8%. For both synthesis, one can notice the use of toxic or CMR reagents or solvent, such as epichlorohydrin, dicholorodiethylether, diethanolamine or toluene.

Ambrozic et al. [132] synthesized a benzoxazine surfactant based on cardanol (BOX, NI-10) in a two-step reaction (Figure 41).

The first step consisted of the partial epoxidation of the unsaturated aliphatic chain of cardanol by H_2_O_2_ in toluene under acidic conditions. About 60% of the alkene functions were converted into epoxy groups. Next, the sequence required a Mannich reaction between the epoxidized cardanol, Jeffamine M-1000 and the toxic compound paraformaldehyde in CHCl_3_ at 61 °C for 20 h in good yield (up to 70%).

NMR analyses of the final product showed that 82% of the BOX molecules have a closed oxazine ring structure.

##### CNSL-Based Surfactant with Polyoxazoline Head

There is only one example [133] of nonionic cardanol surfactant with polyoxazoline, a biocompatible and biodegradable polymer, as hydrophilic head (Figure 42).

Cardanol was esterified with chloroacetyl chloride (a toxic and environmental hazardous reagent) in anhydrous CH_2_Cl_2_ (CMR) in the presence of Et_3_N at 0 °C for 30 min. Chloroacetylated cardanol (CIn) was thus obtained in 98% yield. CIn was used as the macroinitiator for cationing ring-opening polymerization (CROP) of 2-methyl-2-oxazoline (MOx). First, a trans-halogenation on cardanol derivative was carried out by the addition of NaI into a solution of CIn in dry acetonitrile. Then MOx monomer was added to the medium and the polymerization took place at 78 °C for 24 h. COx_m_ surfactants NI-11 with various poly(2-methyl-2-oxazoline) chain length were obtained, with the degree of polymerization ranging from 11 to 28 (M_n_ between 1350 and 2800 g/mol). The biobased part of the surfactant was comprised of between 11 and 22% with decreasing polar chain length.

#### 3.5.2. Highly Biobased CNSL-Based Surfactants

Different strategies have been developed to synthesize this class of nonionic surfactants, starting from cardanol or cardol as the lipophilic group and small sugars (glucose, gluconolactone) or oligosaccharides (amylose, chitosan) as the hydrophilic head.

Two patents described the synthesis of cardanol [134] and cardol [134] derivatives with poly(ethylene oxide) and glucose as polar groups. First, cardanol or cardol were ethoxylated at high temperature in basic medium. Then, ethoxylated cardanol/cardol was reacted with glucose in presence of TiO_2_/SO_2_-4 or TiO_2_-ZrO_2_/SO_2_-4 at 90–120 °C for 2–4 h to afford surfactant NI-12 (Figure 8).

Toxic and petroleum-based ethylene oxide was used in this reaction. The biobased part of such molecules can be varied between 46 and 97% according to the length of PEO chains and/or the number of glucose units. The surface tension properties of such products were not described in the patents.

Prasad et al. [135] also functionalized cardanol with glucose in three steps (Figure 43).

First, unsaturated, and saturated cardanol were etherified with a slight excess of methyl bromoacetate in the presence of K_2_CO_3_ for 8 h. The final products were obtained as white solids. Then, ester groups underwent an aminolysis reaction with hydrated hydrazine (2 eq.) for 12 h. Unsaturated cardanol hydrazide and saturated cardanol hydrazide were obtained with yield = 82% and 93%, respectively. Finally, the hydrazide derivatives were functionalized with glucose in acidic ethanol for 12–24 h. The reaction is proposed to proceed via the formation of hydrazone intermediate, followed by a cyclisation step. 90% biobased glycolipids NI-13 and NI-13′ were obtained with yield = 88 and 92% for unsaturated and saturated compound, respectively. Cardanol-based glycolipids with galactose as hydrophilic group were also prepared by the same process. The use of toxic reagents (methyl bromoacetate and hydrazine) can be noticed in this synthesis strategy.

Cardanol-based surfactant bearing glucosamine as polar group are also described in the literature [136,137] Li et al. synthesized this kind of product according to the following route, described in Figure 44.

The first step consisted of a Williamson reaction in acetone between saturated cardanol and epichlorohydrin (CMR) in basic conditions with TBAB as catalyst. Gluconolactone was meanwhile opened by diethylenetriamine in absolute ethanol at 35 °C. The rection between epoxidized cardanol and amine precursor was carried out in DMSO during 8 h to give 81% biobased NI-14 derivative. Other surfactants were obtained with a similar process, using epoxychlorobutane and/or diethanolamine as reactants.

Cardanol can also be functionalized by longer saccharides, such as chitosan oligomers [138] (Figure 45).

The reaction took place between chitosan oligomers (Degree of Polymerization DP between 5 and 20) and commercially available cardanol glycidyl ether in DMSO at 80 °C for 24 h. Amphiphilic chitosan derivatives of DP 5, 10 and 15 were grafted with cardanol with DS value of 12, 8 and 2%, respectively. These results mean that 40%, 60% and 20%, respectively, of the oligomer chains were functionalized with one cardanol molecule. The authors have filed a European patent describing the synthesis and properties of these surfactants [139].

De Franca et al. [140] prepared CNSL-based surfactants with amylose as polar head (Figure 9):

First, amylose was acetylated and hydrolyzed to obtain acetylated oligomers. Then a glycosylation was carried out between cardanol (NI-16), cardol (NI-17) or anacardic acid (NI-18) and the saccharide compound. The reaction took place in CH_2_Cl_2_ in presence of catalytic amounts of BF_3_.Et_2_O, a CMR compound, for 24 h. Finally, the product was deacetylated in a mixture of 45% trimethylamine/MeOH ¼. 100% biobased glycosides bearing one cardanol, cardol or anacardic acid moiety per oligomer chain were thus obtained. NMR studied showed that phenolic compounds were grafted on the anomeric position of the terminal glucose unit.

## 4. Structure-Property Relationship for CNSL-Based Surfactants

This part of the review aims to provide a better comprehension of the structure-property relationship for CNSL surfactants. Yet, understanding this relationship for these molecules can be quite challenging. Indeed, they are characterized by a large structural diversity, whether in terms of the nature of the polar head (anionic, cationic, zwitterionic or nonionic) and/or the hydrophobic tail into the cardanol, cardol or anacardic acid with the presence of alkenes on the C_15_ side chain combined with other hydrophilic/hydrophobic groups (alcohol, ether, amine, etc.). For example, several publications reported the preparation of Gemini (or dimeric) surfactants, which are composed of two polar heads and two hydrophobic tails, linked together by a spacer [141,142]. Gemini surfactants are known to be considerably more surface-active than conventional (single chain) surfactants. Therefore, a separate section is dedicated to their properties.

Analyses of the structure-property relationship of CNSL surfactants were realized by comparing Critical Micelle Concentration (CMC), surface tension at CMC (γ_CMC_) and sometimes Krafft temperature (T_K_) or Cloud Point (T_C_) of some CNSL-derivatives described in the synthesis section (cf. Figure 10, Figure 11, Figure 12 and Figure 13). Some trends related to the polar head (nature of the hydrophilic group, length) or the alkyl chain (nature of the phenolic compound, unsaturation) are considered here. However, one must bear in mind that experimental data about CNSL-based surfactants are often lacking. For example, very few studies deal with T_K_ or T_C_ of such molecules. As such, it can be difficult to conclude about the influence of a structural parameter on the surfactant properties.

The potency of biobased CNSL surfactants as biodegradable and safer alternatives to commercial products is discussed here. Finally, some (industrial) applications of such molecules are also described.

Anionic CNSL-based surfactants

Four commercial surfactants (Sodium Stearate [143], Sodium Lauryl Sarcosinate [144], SDS and SDBS [145]) were added in this figure (see Figure 10) for comparison purposes.

The CMC value of anionic molecule A-10 (indicated by *) is far higher than CMCs of other cardanol derivatives. The surface properties of A-10 were compared to SDBS in the original publication. The authors also reported a very high CMC of 0.435 mol/L for the commercial surfactant, which is 100 times higher than the data described in the literature. Since CMC values of this research paper seem inconsistent with other published data, it was decided not to take account this result for this review.

Cationic CNSL-based surfactants

Three commercial surfactants (Benzyldimethylhexadecylammonium Chloride [146], Cetyltrimethylammonium Bromide [147,148,149] and Cetylpyridinium Chloride [148,149]) were added in this figure (see Figure 11) for comparison purposes.

Zwitterionic CNSL-based surfactants

Three commercial surfactants (Dodecyldimethyl Betaine [150], Cocamidopropyl Betaine [151] and Lauryldimethylamine N-oxide [152]) were added in this table for comparison purposes.

Nonionic CNSL-based surfactants

Three commercial surfactants (TergitolTM NP-9, TritonTM X-100 and TritonTM CG-600 [153]) were added in this table for comparison purposes.

CMC values of cationic CNSL-based surfactants are globally comprised between 10^−6^ and 10^−4^ mol/L, whereas anionic molecules are characterized by CMC around 10^−3^ mol/L and zwitterionic derivatives with CMC between 10^−5^ and 10^−4^ mol/L. As for nonionic surfactants, their CMC data strongly depend on the nature of the polar head. Indeed, ethoxylated or polyoxazoline derivatives exhibits CMC between 10^−6^ and 10^−5^ mol/L, when sugar-based molecules show CMC values of 10^−3^ mol/L.

### 4.1. Influence of the Polar Head

Anionic CNSL-based surfactants are characterized by a wide diversity of structures. As a result, it is quite hazardous to investigate the influence of the polar head nature (carboxylate or sulfonate group) based on the properties. In the case of cationic derivatives, it is possible to compare the data for homologous series of compounds (Figure 14):

For example, molecules C-14 to C-17 and C-31b only differed by the structure of the quaternary ammonium group: alkyl (trimethyl or triethyl), aromatic (pyridine) or cyclic (morpholino or piperidine) moiety. All surfactants exhibited similar CMC values (around 10^−4^ mol/L), except derivative C-17 (CMC = 2.50 × 10^−5^ mol/L). This molecule possessed the most hydrophobic cationic head, a pyperidinium, which favored aggregation. Globally, CMC values tended to slightly decrease when ammonium heads were substituted with more hydrophobic moieties, such as trimethyl chains or non-aromatic ring. Pyridinium molecule (C-31b) showed the highest CMC (8.00 × 10^−4^ mol/L). According to the authors [104], the cationic charge of pyridinium could be delocalized on the whole aromatic ring, which led to more electrostatic repulsion between such heads.

Another set of homologous surfactants is molecules C-13, C-19 to C-22, C-32, and C-45, where the cationic group is a symmetric or non-symmetric alkyl ammonium (C-19 to C-22), a pyridinium (C-32), a thiazolium (C-45) or a N-methylmorpholinium (C-13). All derivatives showed similar CMC (around 10^−5^ mol/L) and γ_CMC_ (between 13.98 and 22.54 mN/m), except derivative C-13 (CMC = 2.00 × 10^−6^ mol/L and γ_CMC_ = 11.32 mN/m). C_20_ is the concentration of surfactants required to reduce the surface tension by 20 mN/m [142,154], so it characterizes the efficiency of surfactant adsorption. Additionally, the CMC/C_20_ ratio defines the occurrence of either adsorption or micellization process: surfactant characterized by a high value of CMC/C_20_ ratio will preferentially adsorbs at the air/water interface than form micelles. The hydrophobic/hydrophilic nature of the substituents of the ammonium group could also explain these results. For example, molecule C-22, with a hydrophilic hydroxyl function on the polar head, exhibited lower absorption efficiency and micellization tendency than derivative C-21, whose polar head included a hydrophobic linear C_4_ chain. Surfactant C-13 exhibited the highest adsorption efficiency (C_20_ = 1.00 × 10^−6^ mol/L) and the lowest CMC/C_20_ value (2.47). This compound had a strong tendency for micellization in solution, which explained its low CMC.

Some cationic surfactants contain a spacer between the phenol and the polar head. The nature of such a spacer impacts the surface properties. Indeed, for the same polar head, cationic derivatives with hydroxyl function in the spacer (molecules C-19 to C-22 and C-32) exhibit CMC values with a factor of 10 to 100 lower than for derivatives without hydroxyl (C-14 to C-17 and C-31b). The presence of hydroxyl group on the spacer led to intermolecular hydrogen bounds between surfactants, which favored their micellization [155]. Comparison between hydroxyl (molecules C-19 to C-22 and C-32) and unsaturated spacers (C-10, C-43, and C-44) showed that CMC and γ_CMC_ of C-10, C-43 and C-44 derivatives are (slightly) higher than the others. Furthermore, unsaturated surfactants were characterized by higher minimum area per molecule A_min_ than hydroxylated ones. Intermolecular bounding between hydroxylated molecules and/or the higher rigidity of unsaturated surfactants may explain these results.

Regarding zwitterionic surfactants, one study [107] compared the surface properties of pentadecylcyclohexylamine derivatives (resulting from the reductive amination of cardanol) with ammonium and carboxylate (betaine molecule Z-2) or N-oxide (molecule Z-1) as polar head. CMC of betaine molecule was slightly lower than CMC of N-oxide one; yet, both surfactants exhibited CMC values in the same range of magnitude. As for nonionic CNSL derivatives, ethoxylated and polyoxazoline molecules exhibit lower CMC than their analogs with saccharide heads, whereas γ values are of the same order of magnitude (see Figure 13). This phenomenon has been observed for other types of nonionic surfactants [156]: the higher hydrophilicity and steric hindrance of saccharide heads leads to larger CMC.

The number of hydrophilic heads on single hydrophobic tail surfactants influences their behavior. For example, disulfonated cardanol derivative (A-13) is characterized by higher CMC and γ_CMC_ values (1.45 × 10^−2^ mol/L and 60.11 mN/m, respectively) than monosulfonated analog (A-9, 8.35 × 10^−3^ mol/L and 38.41 mN/m, respectively). The same results can be noticed for dicationic molecule C-24 (CMC = 5.62 × 10^−4^ mol/L and γ_CMC_ = 36.88 mN/m) and monocationic C-23 (CMC = 1.38 × 10^−5^ mol/L and γ_CMC_ = 20.17 mN/m). CMC of dicarboxylated surfactant A-4 is also higher than sodium anacardate A-1 and monocarboxylated molecule A-3. However, its γ_CMC_ value is lower than the monocarboxylated analogs. This discrepancy points out the difficulty to determine the impact on the number of hydrophilic head on the surface properties of such surfactants, given the few examples available in the literature. As for zwitterionic surfactants, all are characterized by CMC higher than their one head ionic counterpart. For example, molecules Z-2 (betaine) and Z-1 (N-oxide) showed CMC 20 to 56 times higher than cationic surfactant with similar backbone (molecule C-25). Faye et al. [48] also noticed that zwitterionic derivative Z-3 possessed the highest CMC (1.86 × 10^−4^ mol/L) compared to cationic C-18 (1.37 × 10^−4^ mol/L) and anionic molecule A-20 (7.97 × 10^−5^ mol/L). Indeed, the presence of a supplementary ionic group on the phenolic hydrophobic tail is not in favor of micellization because of repulsive charges between the polar heads. Regarding nonionic CNSL surfactants, Tyman et al. [157] compared the surface tension behavior of 1% aqueous solutions of ethoxylated cardanol (1 polar head, NI-1) and cardol (2 polar heads, NI-2) with degree of ethoxylation comprised between 0 and 48. All molecules reduced surface tension; however, cardol derivatives were less effective than cardanol ones. Atta et al. [131] compared the surface properties of nonionic CNSL derivatives bearing two or three ethoxylated polar heads (molecules DECA NI-8 and TECA NI-9, respectively). Through DLS and interfacial oil/water analyses, the authors showed that DECA molecules were more hydrophobic than TECA derivatives, which explained their lower CMC. A similar study was carried out by Ezzat et al. [109] on cationic cardanol surfactants with two (C-29a) or three (C-29b) PEO chains on the polar head. Cationic derivative with three PEO chains was characterized by a lower CMC and a higher γ_CMC_ than its counterpart with 2 PEO chains. According to the authors, the supplementary PEO arm of cationic C-29b surfactant decreased its solubility in water, therefore lowering its CMC.

Moreover, surfactant behavior is also affected by the length of the hydrophilic group. This point is illustrated by the two series of anionic molecules A-6 and A-6′ (see Figure 10), which comprise a PEO spacer between the phenol and the carboxylate: CMC decreases when the number of PEO units of the spacer increases for each series. This behavior is typical of anionic-nonionic surfactants. As for nonionic surfactants, a slight increase in the CMC with the length of the polar head can be noticed. For example, CMC of ethoxylated cardanol surfactants [158] (NI-1) increased from 5.50 × 10^−6^ mol/L to 2.40 × 10^−5^ mol/L with the degree of ethoxylation (m ranging from 7 and 30). Delage et al. [133] also reported the influence of the length of polyoxazoline chains on the CMC of cardanol-based surfactants (NI-11): CMC values ranged from 4.1 × 10^−6^ mol/L to 1.7 × 10^−5^ mol/L when the degree of polymerization raised from 11 to 28. These data can be explained by the stronger hydrophilicity and steric hindrance of such polar heads, which negatively affects their capacity to self-assemble in water [156]. The Cloud Point of nonionic ethoxylated derivatives is also influenced by the length of the POE head. The Cloud Point is the temperature above which a sample becomes turbid, due to phase-separation between a rich-phase surfactant and the solution. This phenomenon is due to PEO chains, whose solubility in water decreases when the temperature is raised [159]. Cloud Point values of nonionic CNSL derivatives NI-1 rose from 28 °C to 82 °C with the number of ethylene oxide units in the polar head (from m = 5 to m = 10) [121]. A similar result was described by Wang et al. [158] for ethoxylated molecules with m comprised between 7 and 30.

Surface properties of ionic CNSL derivatives are influenced by salt concentration [48,85,160]. CMC values decrease with increasing electrolyte concentration, as illustrated for anionic surfactant A-13 (see Figure 10). Indeed, adding salt reduces electrostatic repulsion between ionized polar groups, therefore favoring micellization. Kraft temperature T_K_ can also be affected by the presence of salt, as illustrated by Table 1.

One can notice a small decrease in T_K_ for cationic surfactant C-18 when NaCl concentration was raised from 0 to 500 mmol/L. The evolution of T_K_ with NaCl was more pronounced for anionic A-20 and zwitterionic Z-3 derivatives. First, these two molecules were not soluble in water, contrary to cationic C-18 surfactant. Increasing salt concentration favored their solubility in water, which explained the decrease in their Kraft temperature.

### 4.2. Influence of the Alkyl Chain

Probing the influence of the hydrophobic tail nature (cardanol, cardol or anacardic acid) on the amphiphilic properties of CNSL-based surfactants can be quite challenging. Indeed, most derivatives described in the literature are synthesized with cardanol as the hydrophobic group. For example, no example reported the preparation of cationic surfactants from anacardic acid and cardol. To the best of our knowledge, only one study [74] compared the amphiphilic behavior of carboxylated anacardic or cardanol-based surfactants (molecules A-1, A-2, and A-3, Figure 10). Anacardic derivatives showed slightly higher CMC and γ_CMC_ values than cardanol. Both CMC were quite close, but γ_sodium anacardate_ (35.10 mN/m) was higher than γ_triethanolamine anacardate_ (30.34 mN/m). The authors used the hydrophilic-lipophilic deviation (HLD) index [160,161] to determine the hydrophilicity of their surfactant. This criterion can be quantified by σ, which is a characteristic parameter of the molecule. It can be positive or negative: the higher the value, the more hydrophobic the surfactant. Cardanol acetate (A-3) was the more hydrophobic molecule (σ > 0), which explained its lowest CMC and γ_CMC_ results. On the contrary, triethanolamine anacardate (A-2) showed the lowest σ value (σ = −5.4), followed by sodium anacardate (A-1, σ = −2.3). Therefore, triethanolamine salts were more hydrophilic compared to sodium anacardate because the bulky amine counterion prevented the formation of intramolecular hydrogen bound between the phenol and the acid group. As a result, the acid function was more prone to ionization than its sodium analog, which explained its more surface-active properties.

Regarding nonionic CNSL surfactants, a complete study of the impact of the phenolic hydrophobic tail on the molecule properties was realized by de Franca et al. [140]. They synthetized nonionic surfactants with amylose as polar head and cardanol, cardol and anacardic acid as hydrophobic group (NI-16, NI-17, NI-18, Figure 13). CMC, surface excess Γ, and the area per molecule A, were determined for each surfactant. CMC of each product were of the same order of magnitude, but it was noticed that CMC_cardanol_ < CMC_cardol_ < CMC_anacardic acid_. In the case of anacardic acid, the presence of an ionizable, carboxylic function on the phenolic ring led to a lesser packaging at the air/water interface, thus decreasing its surfactant properties. The relationship between structure and surfactant properties of cardanol and cardol derivatives was more ambiguous. Indeed, it was observed that A_cardol_ < A_cardanol_, even though CMC_cardol_ > CMC_cardanol_. As stated by the authors, cardol glycosides formed smaller aggregates in solution than its cardanol counterparts due to the supplementary phenol group, which favored the formation of intermolecular hydrogen bounds between surfactants. However, the discrepancy in CMC values between these two CNSL derivatives was not explained. The measure of the critical packing parameter (cpp) indicated that the CNSL surfactants formed vesicle and bilayer aggregates in solution. To conclude, the nature of the phenolic compound (cardanol, cardol or anacardic acid) can have a moderate impact on the surfactant properties of CNSL derivatives. The occurrence of intramolecular (for anacardic acid) or intermolecular (for cardol) hydrogen bounds seems to be the main factor governing the surface properties. Further experimental investigations should be encouraged to confirm (or not) this trend.

To the best of our knowledge, there are three examples of ionic CNSL-based surfactants with 5-pentadecylcyclohexane as hydrophobic tail (Figure 15). This moiety is obtained by reductive amination of CNSL.

CMC of such compounds are lower than those previously given for cardanol-derived surfactants with aromatic group and commercial products. The variation in hydrophobic chain by the presence of a cyclohexane part clearly influences the micellization process, albeit for unknown reasons. A more complete study of the surface properties of such derivatives should be encouraged to better understand this peculiar behavior.

Very few experimental studies about the influence of unsaturation on the surfactant properties of CNSL derivatives are available in the literature. Carboxylated cardanol-based surfactant, with and without unsaturation, were synthesized by Wang [78] and Li [80], respectively, (molecules A-6 and A-6′). For each molecule, a spacer PEO group was introduced between the phenol and the polar group. CMC of saturated compounds were lower than CMC of unsaturated ones, whatever the length of the spacer. Lower CMC values for saturated products compared to unsaturated ones have already been reported in the literature for other surfactants [156]. These results could be explained by the steric hindrance generated by double bounds during the packing of unsaturated surfactants at the air/water interface; and by their better solvation in water compared to alkyl chains. On the other hand, surface tension values of saturated surfactants were higher than unsaturated ones, especially for molecules with a spacer arm composed of 10 PEO units (γ_saturated C15 chain_ = 43.53 mN/m, whereas γ_unsaturated C15 chain_ = 28.60 mN/m). No explanation was available regarding these results. Wang et al. [76] studied the surface activity, at 45 °C and pH = 12, of unsaturated and saturated benzoxazine surfactants based on cardanol (molecules A-5 and A-5′, respectively). Both molecules were characterized by almost identical CMC and surface tension data. The authors noticed that the minimum area per molecule A_min_ was slightly lower for saturated benzoxamine (A_min_ = 72.8 Å2) than for unsaturated one (A_min_ = 77.9 Å2). Unsaturated derivatives had a lower packing density than that of saturated ones at the air/water interface, due to the steric hindrance of the double bound. Nevertheless, the presence of double bounds on the side chain does not seem to impact the surfactant properties of these derivatives.

To the best of our knowledge, only two publications [135,157] reported the impact of unsaturation on the behavior of nonionic CNSL derivatives. Tyman et al. [157] measured the surface tension of 1% aqueous solutions of cardanol and 3-pentadecyl polyethoxylate (m between 0 and 48). All ethoxylated derivatives showed similar behavior in reducing surface tension. The best results were obtained for cardanol molecule with m = 13.4 and 3-pentadecyl molecule with m = 14.8: both were characterized with γ_min_ ≈ 37 mN/m. Ethoxylated unsaturated cardanol exhibited slightly better properties than its saturated analog. Prasad et al. [135] reported that glycolipids synthetized with unsaturated (NI-13) or saturated (NI-13′) cardanol hydrophobic tail had the same CMC, but γ_saturated cardanol_ (44 mN/m) was lower than γ_unsaturated cardanol_ (51 mN/m). In both examples, the presence of double bounds in the hydrophobic tail of CNSL surfactants decreases the surface activity of such molecules, but on a small scale. To sum up, the few reports about the effects of unsaturation of CNSL-based molecules on their surface properties are inconclusive. Furthermore, the lack of data regarding this structural parameter is important. More studies on this subject would be appreciated to determine its real influence on surface behavior.

### 4.3. Gemini Surfactants

As previously mentioned, some examples of CNSL derivatives are Gemini (or dimeric) surfactants. They are composed of two polar heads and two hydrophobic tails that are linked together by a spacer [141,142]. For anionic Gemini molecules, the hydrophilic polar group can be a carboxylate [80] or a sulfonate [83,162] and an alkyl ammonium [101,103], a pyridinium [111]/bipyridinium [163], or an imidazolium [111] for cationic derivatives. To the best of our knowledge, one example [126] of nonionic Gemini surfactants with PEO as polar groups are reported in the literature (molecule NI-3), yet their surface properties were not studied. All Gemini surfactants are based on cardanol as the hydrophobic tail. Most of them are symmetric (two identical hydrophobic tails), but two examples of cationic Gemini are non-symmetric (molecules C-11 and C-12): one hydrophobic tail is cardanol, the other is a linear alkyl C_4_ or C_6_ chain. Surface properties of some CNSL-based Gemini surfactants are summarized in Figure 16 and Figure 17.

CMC values of anionic Gemini CNSL-based surfactants are comprised between 6.20 × 10^−5^ and 1.9 × 10^−3^ mol/L, and cationic Gemini molecules between 5.40 × 10^−6^ and 3.00 × 10^−4^ mol/L. These derivatives have much lower CMC than commercial products, such as SDBS or cetyltrimethylammonium chloride. These data agree with those reported for Gemini surfactants [164,165]. Anionic Gemini molecules exhibit lower CMC than their conventional ionic analogs. For example, CMC of Gemini molecule A-8 is 7.90 × 10^−4^ mol/L (for n = 8), whereas CMC of its counterpart A-6′ is 1.45 × 10^−3^ mol/L. For cationic Gemini surfactants, only one study [101] compared the CMC data between a dimeric molecule (C-4) and its single-chain counterpart (molecule C-3b): both surfactants had the same CMC. Yet, two series of pyridinium and imidazolium surfactants with a sulfur spacer (molecules C-33 to C-36 and C-38 to C-41) undergo self-aggregation at very low concentration ranging from 5.40 × 10^−6^ to 2.09 × 10^−5^ mol/L. These data are quite low compared to cationic Gemini surfactants with aliphatic hydrophobic tail [111,165]. Indeed, the presence of a rigid, hydrophobic benzene ring between the alkyl chain and the ionic head favors micellization at lower concentration [166].

Notably, γ_CMC_ of anionic molecules are of the same order of magnitude than commercial surfactants (Figure 16), whereas γ_CMC_ of cationic compounds are lower (see Figure 17). Surface properties of Gemini surfactants can also be characterized by the efficiency parameter C_20._ For example, the efficiency factor of anionic Gemini molecule A-16 was C_20_ = 4.40 × 10^−5^ mol/L, whereas its single chain analog (A-9) was C_20_ = 2.80 × 10^−4^ mol/L. Shi et al. [162] compared the efficiency of anionic Gemini derivative A-17 and a single-chain cardanol sulfonate: the former showed a higher efficiency at reducing surface tension (C_20_ = 6.00 × 10^−5^ mol/L) than the latter (C_20_ = 4.50 × 10^−3^ mol/L). By comparison, commercial products such as SDS and SDBS are characterized by C_20_ of 2.04 × 10^−3^ mol/L [167] and 1.99 × 10^−4^ mol/L [168], respectively. As for cationic Gemini derivatives, one study [103] reported the C_20_ data of cationic Gemini with 1,4-diazabicyclo[2.2.2]octane as part of the spacer (molecules C-11 and C-12): C_20_ of derivatives C-11 and C-12 were 5.40 × 10^−6^ mol/L and 4.00 × 10^−6^ mol/L, respectively. Commercial surfactants CTAB and cetylpyridinium chloride show C_20_ values of 4.07 × 10^−4^ mol/L and 3.55 × 10^−4^ mol/L, respectively [149]. Gemini CNSL-based derivatives are, therefore, a great and efficient alternative to petroleum-based surfactants.

It is well known that the type of spacer affects Gemini surface behavior. Wang et al. [101] synthesized cationic Gemini surfactants with hydrophobic flexible or rigid spacers (molecules C-3b and C-5 to C-7). Gemini derivatives with linear, flexible aliphatic chain as spacer (C-6 and C-7) showed lower CMC than molecule with a rigid aromatic spacer (C-5), in agreement with the literature [154,169,170]. However, it was observed that cationic surfactant bearing benzophenone spacer (C-4) exhibited the lowest CMC values, even though its spacer was the more rigid. No explanation was given by the authors about this result. The influence of the spacer length is more unclear. Bhadani et al. [111] compared the properties of cationic Gemini surfactants having sulfur spacers of different length between the hydrophobic tails (molecules C-33 to C-36 and C-38 to C-41). CMC values decreased with increase in spacer length, as reported for other Gemini derivatives [170]. However, in the case of anionic Gemini derivatives A-18, no clear trend can be observed between the length of the spacer and the surface behavior.

All CNSL-based Gemini surfactants described in the literature have been produced with cardanol as the hydrophobic tail. Therefore, it is not possible to study the influence of the phenolic compound (anacardic acid, cardol, cardanol) on the surface properties of such molecules. One study [103] described the synthesis of dissymmetric Gemini surfactants with cardanol as one hydrophobic tail and linear C_4_ or C_6_ chain as the second tail (molecules C-11 and C-12). Both derivatives possessed similar CMC values, which were in the same order of magnitude than other cationic Gemini surfactants. It was observed that Krafft temperature (T_K_) of molecule C-12 with C_6_ tail (T_K_ > 90 °C) was much higher than those of molecule C-11 with C_4_ tail (T_K_ = 3 °C). Increasing the length of one hydrophobic tail led to a decrease in the solubility of such compound in water, which explains the higher T_K_ of derivative C-12.

Some applications of such Gemini surfactants are depicted here. First, three Chinese patents [81,94,126] about the synthesis of anionic (molecules A-8 and A-18) or nonionic (NI-3) cardanol derivatives can be found in the literature. This fact points out the interest of such compounds by the industrial actors as biobased alternatives of petroleum-products. For example, Hu et al. [126] characterized the cloud point of nonionic Gemini surfactants (NI-3) with polar head length between 6 and 60 PEO units. Except for the molecule with rigid aniline as spacer, which possessed a low cloud point equal to 36 °C, all surfactants exhibited cloud points between 62 and >100 °C. Zhou et al. [94] showed that anionic Gemini molecule A-18 exhibited interfacial crude oil/water properties and, therefore, could be used for enhanced oil recovery. Ahire et al. [83] studied the foamability of their anionic Gemini surfactants (A-16) and their single chain counterpart (molecule A-9). Gemini derivatives possessed low foaming capabilities compared to their analog; yet, they could be considered as low foam producing surfactants for application in washing machine laundry, spray cleaners or oilfield additives. Cationic Gemini molecules C-4 to C-7 were employed as emulsifier for emulsion radical polymerization of methylmethacrylate with AIBN as initiator. Poly(methyl methacrylate) with monomer conversion equal to 85% (70 °C polymerization during 12 h) and molar mass around 85,000 g/mol was obtained [101].

Some Gemini CNSL derivatives exhibited interesting antimicrobial activities. For example, anionic molecules A-16 showed favorable antimicrobial properties against both Gram-positive and Gram-negative bacteria, such as S. aureus or E. coli. These surfactants were also efficient against fungi such as A. niger, C. albicans or A. flavus. The authors [83] concluded that these Gemini CNSL derivatives could be used as antimicrobial agents against plant, animal, and human pathogens. Non-symmetric cationic Gemini molecules C-11 and C-12 also presented antibacterial activities against Gram-positive bacteria B. subtilis and Gram-negative bacteria E. coli [103]. Finally, cationic derivatives C-33 to C-35 were able to bind to DNA, with imidazolium molecules showing greater binding capabilities than pyridinium surfactants [111]. Moreover, cytotoxicity evaluation of these molecules on cancerous brain cells proved that they possessed low toxicity compared to commercially available transfecting agent dimethyldioctadecylammonium bromide. As a result, these cationic Gemini CNSL-based surfactants could be used as gene delivery agents.

### 4.4. Biological Activity, Cytotoxicity, and Biodegradability

Its natural abundance and the biological activity of the molecules that compose it are undeniable strengths of CNSL. Anacardic acid is well known as an antibacterial and antifungal agent [67,171,172]. It is a particularly interesting active property in the field of detergency and soaps. Several CNSL-based surfactants retained these biological properties. This was the case of the diacid A-4, an anionic surfactant, described by Khatib et al. [75] which acted as an antibacterial on S. aureus, E. faecalis, K. Pneumoniae, E Coli and P. aeruginosa. Ahire et al. [83] described a Gemini sulfonate surfactant that also exhibited antifungal and antibacterial properties. Finally, several Chinese patents [124,173] detailed the use of antibacterial nonionic surfactant from ethoxylated CNSL for the synthesis of hand sanitizer.

However, there is a trade-off between biological activity and cytotoxicity. Most surfactants will be used in contact with skin and hair, or end up in an aquatic environment. Despite the efforts of some authors to demonstrate the non-cytotoxicity of their molecules [76,111,174] or their non-irritating aspects [173], this information is, unfortunately, too often lacking in published articles. In Bhadani et al. paper [111], it was highlighted that the linker or polar head (here the imidazolium head is less cytotoxic than the pyridinium head) can have a significant impact on surfactant toxicity. This makes it even more important to systematically evaluate the potential cytotoxicity of the synthesized product. A bio-based molecule is not automatically a non-hazardous one. Many examples in nature easily prove otherwise. Finally, Jorge et al. [174] showed that the functionalization of CNSL, here by sulfonation, can also negatively impact the original biological activity of the synthon, transforming there a larvicide into a simple non-cytotoxic surfactant.

The synthesis of zwitterionic copolymers Z-4 with cardanol and a significant proportion of sulfobetaine core not only revealed excellent bactericidal properties, but also an interesting adhesion resistance to proteins, bacteria and cells, with sufficient biocompatibility [113]. When the copolymers contain more than 58 mol% of cardanol, the bactericidal propriety increases against Escherichia coli. Moreover, with a copolymer containing 42 mol% of sulfobetaine moiety, excellent biocidal and antifouling properties and sufficient biocompatibility was observed. These results are due to a hydration layer linked to a strong interaction between the amphiphilic moiety and water molecules.

The end of life of a product is also very important. CNSL and its compounds are biodegradable even if there are, to our knowledge, few articles on this subject. Unfortunately, little information or study about the biodegradability of surfactants derived from CNSL are available, even though these products are often relegated to our ecosystem after use. However, a particular interest in ethoxylated compounds can be noted. Indeed, several studies showed and confirmed the biodegradability of these derivatives. Tyman et al. [157] studied the biodegradability, by soil microorganisms, of saturated and insaturated cardanol and cardol with 13 and 10 units of PEO, respectively. After 28 days, the remaining Total Organic Carbon was estimated at 17% and 25% for cardanol polyethoxylate and its saturated analog, as well as 37% and 46% for cardol polyethoxylate and its saturated analog, respectively. In comparison, t-nonylphenyl polyethoxylate remained essentially undegraded (77%) and today, in the process of substitution. The difference between cardanol and cardol compounds was explained by the presence of two ethoxylated arms on the cardol compared of one for cardanol, which could slow their biodegradation. A patent [121] also investigated the biodegradability of polyethoxylated cardanol with m between 1 and 30 by a cobalt thiocyanate method: about 100% of the cardanol derivatives was degraded after 7 days. Moreover, ethoxylated cardanol derivatives are less irritating for skin than petroleum-based surfactant [124,173]. However, their synthesis required the use of ethylene oxide, a very toxic precursor.

### 4.5. Comparison between CNSL-Based and Conventional Surfactants

To better understand the potential of CNSL-based surfactants for industrial applications, we compared literature values of CMC, γ_CMC_ and, when available, C_20_ and/or T_K_ of ionic and nonionic single chain CNSL derivatives to commercial products.

Carboxylated CNSL-based surfactants are characterized by CMC between 8.30 × 10^−4^ and 8.10 × 10^−3^ mol/L. These values are close to CMC of commercial carboxylated products such as sodium stearate and SLSA. Surface tension values are ranging from 28.38 to 43.53 mN/m, which are of the same order of magnitude than commercial ones. Moreover, some carboxylated molecules (A-1 and A-2) are characterized by T_K_ lower than 5 °C, which make them interesting for industrial processes requiring low temperature. As for sulfonated CNSL derivatives, their CMC are comprised respectively between 1.70 × 10^−5^ and 8.35 × 10^−3^ mol/L (except molecule A-13, with a high CMC of 1.45 × 10^−2^ mol/L), which are lower than SDS and SDBS. We noted a surprising and unlikely CMC value for A-9 with 372 mmol/L [82]. As previously mentioned, it is probably an error from this article. Their γ_CMC_ values range from 31 to 42 mN/m, such as commercial anionic products. Super spreadability or super wettability aromatic siloxane sulfonated surfactants A-14 and A-15 were characterized with low γ_CMC_ values of 18–24 mN/m. Therefore, anionic CNSL-based surfactants can be as effective as commercial ones. To the best of our knowledge, many patents report the synthesis and/or applications of anionic CNSL-based surfactants, which show their industrial potential as biobased alternatives to petroleum products.

Cationic CNSL surfactants exhibit CMC between 5.00 × 10^−7^ and 8.00 × 10^−4^ mol/L and γ_CMC_ varying from 11.32 to 43.2 mN/m, whatever the nature of the polar head (alkyl ammonium, pyridinium, thiazole or N-methylmorpholine). Globally, their surface parameters are much lower than commercial cationic surfactants (Figure 18).

CMC of CNSL derivatives are 10 to 100 times lower than those of commercial products such as CTAB or CPC. Moreover, CNSL molecules are more efficient than commercial ones to reduce the surface tension at the air/water interface, as indicated by their γ_CMC_ and C_20_ values. They also exhibit lower T_K_ that expands their application field compared to conventional cationic surfactants [146,175,176]. Many examples of cationic CNSL surfactants are described in the literature, yet there are only two patents about the synthesis of such derivatives for industrial purposes. Biobased cationic CNSL products show interesting antibacterial properties; therefore, they could replace petroleum compounds for such applications.

Zwitterionic surfactants derived from cardanol are poorly developed and there is not much information on their physico-chemical properties. It can noticed, however, that around 40% of the available literature about such products are patents.

The surface tension and aggregation behavior of zwitterionic surfactants Z-1 and Z-2 were determined using drop shape analysis. The critical micellar concentration was found to be 2.80 × 10^−5^ and 1.00 × 10^−5^ mol/L for these two compounds, respectively. CMC of such compounds are lower than common commercial zwitterionic products, such as cocamidopropyl betaine or LDAO.

Notably, γ_CMC_ and T_K_ were described in the literature only for molecule Z-3. Due to the low solubility of this compound in water, sodium chloride concentration has a significant influence on the Krafft temperature which decreases from 13 °C to <0 °C when the concentration of NaCl increased from 50 mM to 500 mM. Furthermore, γ_CMC_ and T_K_ were estimated at 30.29 mN/m (close to lauryl betaine and LDAO) and <0 °C, respectively. Therefore, this CNSL surfactant shows promising potential as an alternative to commercial zwitterionic products.

Ethoxylated and polyoxazoline CNSL surfactants exhibit CMC ranging from 10^−6^ mol/L to 10^−4^ mol/L. These values are lower or close to the commercial ones (Tergitol^TM^ NP-9 and Triton^TM^ X-100). Additionally, γ are generally comprised between 32 (close to commercial products) and 53 mN/m, with no clear trend between the surface tension and the surfactant structure. Some of these biobased molecules can replace petroleum-based commercial surfactants, such as Tergitol^TM^ NP-9. CNSL derivatives with saccharide head show higher CMC than their analogs with PEO or polyoxazoline hydrophilic groups, whereas γ values are of the same order of magnitude. Comparison between CNSL-based and commercial saccharide surfactants (such as Triton^TM^ CG-600) is tricky, because of a lack of data regarding CMC values in the literature. It can be noticed that CNSL derivatives with glucose or glucosamine are characterized by much larger CMC (between 1100 and 4100 mg/L) than Triton^TM^ CG-600 (CMC = 74 mg/L), which seems to point out that these molecules are less effective than the commercial one. Yet, CNSL derivatives with amylose as polar head exhibit CMC (in mol/L) and γ close to commercial Triton^TM^ X-100. The latter was classed in the Authorization List (Annex XIV) of the Registration, Evaluation, Authorization and Restriction of Chemicals (REACH) because of one ecotoxic degradation product. CNSL-based molecules with amylose head can, therefore, be an interesting alternative to such surfactants. More than 20 patterns report the synthesis and/or application of nonionic CNSL-based surfactants, especially ethoxylated molecules. Moreover, some of them are sold by Cardolite or K2p Chemicals. This fact points out the interest of such compounds by the industrial actors as biobased, non-toxic, and biodegradable alternatives of petroleum-products such as nonylphenol ethoxylate surfactants.

### 4.6. Applications of CNSL-Based Surfactants

CNSL-bases surfactants have found various uses in many fields such as detergency, polymerization emulsion, enhanced oil recovery or antimicrobial products. Some applications of these compounds are described in this section.

CNSL surfactants can be employed in formulation of detergency solutions [118,124,158,177,178,179]. For example, cationic derivatives C-19 to C-24 and C-32 showed good emulsifiability and low surface tension, which facilitated foamability. In combination with a non-ionic surfactant, a cardanol polyoxyethylene ether, and an anionic surfactant, SDS, the cationic surfactant with hydroxyl function and triethylammonium group C-20 allowed the preparation of detergents. Washing tests were performed with dirty clothes. The formulation prepared with the cationic surfactant had better detergency than formulation with only anionic and non-ionic species. A pilot scale-up production of 20 kg detergent was carried out. In addition, these plant-based surfactants could be a greener substitute for TX-10, a nonylphenol polyoxyethylene ether which is a product of petrochemical derivative. Jin et al. [178] prepared oil-in-water emulsions containing nonionic ethoxylated cardanol (NI-4). They recorded the time required to observe a definite separated volume of water: the higher the time, the stronger the emulsification capability. Cardanol derivatives were able to stabilize such emulsions in the same range of time than commercial alkylphenol polyethoxylated surfactants (such as Tergitol^TM^ NP-10). A washing test on soiled cotton fabric proved that NI-4 has similar performances to petroleum-based Tergitol^TM^ NP-10. Similar results were described by Zhu [179] or Wang [158] in comparison to standard laundry formulations.

Nonionic ethoxylated CNSL surfactants were also used as surfactant in cement formulation [180] or in hand sanitizer [173]. Furthermore, Chapelle et al. [138] showed that direct oil-in-water emulsions (60/40), stable for 24–48 h and with 15 μm medium size droplets, can be obtained with cardanol surfactants with chitosan oligomer as polar head (NI-15). They showed promising potential as new biobased emulsifier for bitumen emulsions [139].

Another example of the application of CNSL derivatives is their use as stabilizer, especially for polymerization emulsion. The carboxylated surfactant A-5 based on benzoxazine ring was employed for the preparation of polystyrene latex. It offered high efficiency for the stabilization of emulsion. Anionic surfactant A-19a, with n = 1 and 2, were successfully utilized by Jayakannan et al. [181] as structure directing agents for polyaniline nanomaterials-nanofibers and nanotapes by emulsion and dispersion routes. These sulfonic surfactants allow the synthesis of high-quality nanofibers, due to spherical or cylindrical micellar self-organization and open perspectives to chemical sensors and optical devices. Otherwise, A-19b (n = 2) was also incorporated into epoxy resins by Wang et al. [182] after modified layered double hydroxide. This combined additive improves the flame retardant properties with a better dispersion of nanofiller in the matrix. Cationic surfactant C-3a (with a benzyl bromide group) could act as both an AGET ATRP (Activator Generated by Electron Transfer Atom Transfer Radical Polymerization) initiator and an emulsifier for emulsion polymerization. MMA polymerization was performed with the catalytic system based on CuBr_2_ and bipyridine. To obtain a better emulsion stability, a mixture of these two surfactants was used and the polymerization afforded a stable latex. According to Caillol et al. [25], zwitterions derived from cardanol such as surfactant Z-3 could be used in emulsion polymerization for the synthesis of styrene-acrylate latex. Yet the study of such properties was not performed. As for nonionic derivatives, Ambrozic et al. [132] used a benzoxazine surfactant based on cardanol (BOX, NI-10) as stabilizer for epoxy aqueous emulsions. Such molecules were able to stabilize emulsions containing synthetic epoxy resin (such as bisphenol A diglycidyl ether) and/or bio-renewable epoxidized soybean oil. After drying and epoxy resin curing, benzoxazine surfactant became an integrated part of the epoxy-benzoxazine copolymer network and improved its thermomechanical properties.

Some CNSL surfactants were employed as demulsifier agents [109,130,131] in crude oil emulsions. For example, the macromolecular surfactant (C-29) was used to disperse the asphaltene fractions of the heavy Arabian crude oil. Indeed, this amphiphilic poly(ionic liquid) has the ability to reduce the interfacial tension and can separate water effectively. This surfactant achieved separation performance and demulsifying action reached 100% with a low concentration of 10 mg/L for crude oil/water (90/10 vol %) emulsion. Atta [131] and Feitosa [130] studied the demulsifying capability of their nonionic ethoxylated cardanol products (molecules NI-7, NI-8 and NI-9). The former observed total demulsification for Arabic crude oil/water emulsions (90/10 vol %) formulated with 50 mg/L of DECA. On the other hand, TECA showed high demulsifying capabilities (at concentration ≥ 100 mg/L) for crude oi/water emulsions with higher water contents (50/50 vol %). These results were explained by the difference in hydrophobicity of each surfactant [131]. Feitosa et al. [130] prepared water-in-Brazilian crude oil emulsions (30/70) with ethoxylated cardanol (NI-7) and ethoxylated cardanol formaldehyde resin as surfactants. Better demulsifying properties were obtained for ethoxylated cardanol molecules. At last, cardanol betaine surfactants Z-5 or Z-6 were used, alone or in combination with other additives, as oil displacement agents. These derivatives showed good interfacial properties with crude oil even at high temperature (85 °C) and high salinity (40 g/L), which made them suitable for waterflooding applications in petroleum enhanced recovery field.

Notably, Z-7 and Z-8, polyaromatic cardanol zwitterionic surfactants have demonstrated good temperature resistance, salt resistance and interface activity. A water solution of these cardanol zwitterionic surfactants can have ultralow interfacial tension with crude oil, so that the cardanol zwitterionic surfactant has a good application prospect in several oil reservoirs with high temperatures and high mineralization degrees.

Several studies reported the antibacterial properties of cationic surfactants obtained from cardanol. The minimum inhibitory concentration (MIC) and minimum bactericidal concentration (MBC) of cardanol surfactants with ammonium and thiazolium groups (C-10, C-13 C-43 to C-45) were determined against *S. aureus*, *B. subtilis* and *E. coli* [103]. It was observed that surfactants C-10, C-13 and C-43 were superior to traditional single surfactants; they can inhibit or even kill the tested bacteria at a concentration equal to 32 μg/mL. As explained in most of the literature, the cationic molecules adsorb on the cell membranes by electrostatic attraction and interact with bacterial cell membranes leading to disintegration. In addition, a rise in polarity of the head group (for example by adding double bond and hydroxyl groups) can promoted antimicrobial properties.

Mhaske et al. [110] have studied the preparation of a methacrylate monomer derived from cardanol with quaternary ammonium groups and brominated functions C-30. This monomer was incorporated in UV curable coatings by mixing with epoxy acrylate oligomer and photoinitiator. Antimicrobial performances of the film were studied with different microorganisms as bacteria, yeast, and fungi. Compared with a film without cardanol-derived monomer, the antibacterial performance was increased with a significant inhibition percentage value of 81.59% for gram-positive bacteria (*S. aureus*), 77.12% for gram-negative bacteria (*E. coli*) and 77.82% for yeast (*Candida albicans*). Cationic surfactants (C-8, C-9 and C-31a) can be used to disperse nanomaterials as single-walled carbon nanotubes in aqueous solution, because benzene rings and aliphatic chains enable strong π-stacking interaction on carbon nanotubes. Surfactants prepared from cardanol by introduction of ammonium or pyridinium groups are better for carbon nanotubes dispersion than the classical anionic surfactant, dodecylbenzene sulfonate sodium (SDBS). In addition, carbon nanotubes dispersed with cardanol-derived surfactants demonstrate significantly improved antibacterial properties against E. coli and S. aureus. The antibacterial activities are attributed to positive charges introduced by ammonium and pyridinium groups. In the specific case of zwitterionic copolymers Z-4 containing cardanol and zwitterionic groups, Kim et al. [113] indicated that promising coating materials can be made from these polymers in biomedical devices and water purification. As for nonionic CNSL surfactants, Prasad et al. [135] showed that cardanol-based glycolipids NI-13 and NI-13′ were able to disrupt biofilms formed by pathogens, such as E. coli and Salmonella enterica Typhimurium. These biobased products could be used for surface cleaning in hospital environments or food processing industries.

## 5. Perspective and Conclusions

CNSL is a generous raw material with high potential. Generating many synthons in the field of polymers and additives; their application in the field of surfactants is no exception to the rule. From “A Brief Review of Cardanol-Based Surfactant” by Yang et al. in 2014 [24], the number of publications using one of the CNSL compounds for surfactant synthesis increased significantly. Numerous patents attest to the enthusiasm and interest of industries on this bio-based solution. It is also mainly patented from the Asian continent, which is very active in the production and valorization of these co-products of the cashew industry.

Considering a little more closely, the nature of the surfactants obtained or their synthesis, it can be noticed that, for the moment, it is far from an idyllic model. Even if the physicochemical properties (CMC, surface tension) of these surfactants compete with the commercial products currently on sale and in the process of substitution, many of these surfactants are biobased only from the hydrophobic part of their structure provided by the CNSL derivative. Indeed, except for some nonionic structures, where the polar head is a derivative of sugar, most come from compounds that are petrochemicals and/or use toxic/CMR reagents such as sultones or ethylene oxide.

Many synthetic pathways also use reagents from conventional chemistry, which are also dangerous for humans and the environment. It would be good to change this—to respect green chemistry. Finally, despite the interest and growing importance of producing non-toxic and ecologically friendly molecules, it is regrettable that little study has been carried out on the possible intrinsic toxicity of synthesized surfactants, despite the known problems of certain structures such as quaternary ammoniums or sulfonates, to name but a few. Some still make the effort to show the biodegradable behavior of their molecule.

In the many examples of surfactants synthesized from CNSL present in this review, it can easily be noticed that most of them are cardanol derivatives. Despite the natural biological activity of anacardic acid, it is often little exploited. The reason probably comes from the CNSL extraction method. Anacardic acid, corrosive, is often decarboxylated during extraction to facilitate the recovery of oil from the cashew shell. However, anacardic acid, such as cardol, can be the source of interesting innovations in the field of surfactants, whether in detergent or cosmetic, and will most certainly in the future be more prevalent and used for surfactant synthesis.

It should also be noted that the available information on the properties of synthesized surfactant structures is very disparate from one article to another. Even if, in general, the values of CMC and the surface tension of molecules can be found, other information such as Kraft temperature or Cloud Point is regularly missing. Indeed, many of the articles cited in this review, and more often patents, focus on an application of the synthesized surfactant, omitting at the same time other relevant data that it would have been good to share with the scientific community. Nor will we mention here the particular difficulty of finding or calculating hydrophilic-hydrophobic balance (HLB) values—data that are nevertheless useful for industry, which differ according to the nature of polar heads and where, for lack of data on certain chemical structures, it is not even computable.

CNSL is, therefore, a good track as synthons for the synthesis of effective surfactants. Much progress has been made over the last 10 years and efforts must be maintained to synthesize and produce non-hazardous surfactants that are the most biobased as possible. Synthesis routes still need to be optimized considering the fundamental principles of green chemistry. The future of this resource will be shaped by simple, inexpensive, and virtuous chemistry, involving the use of all CNSL compounds or even crude oil itself. CNSL provides the hydrophobic part of the surfactant, so it is necessary to imagine, as it is already the case for some nonionic surfactant, the use of bio-based polar heads, which also meet the criteria of non-toxicity. We can then think of sugars, glycerol already used but also some amines, amino acids, acids and diacids, etc.

Other interesting perspectives concern CNSL derivatives that are, for the moment, not described as surfactants, but are already synthesized, and whose structure is challenging. Indeed, this is the case of some polyols [183,184] derived from cardanol for the synthesis of polyurethane resins, whose polarity could potentially be sufficient to induce surfactant properties. Finally, CNSL can also, more anecdotally, intervene in the production of biosurfactant, which are, nowadays, increasingly studied and described [185,186,187]. Satheesh et al. [188] described, for example, the production of a biosurfactant by Pseudomonas sp, fed with CNSL.

CNSL, a promising resource, has not yet revealed all these possibilities and therefore remains a relevant molecule to be followed in the synthesis of tomorrow’s bio-based surfactants.

## Data Availability

Not applicable.

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
