# Peer review of "CNSL, a Promising Building Blocks for Sustainable Molecular Design of Surfactants: A Critical Review"

_molecules, 2022, doi:10.3390/molecules27041443_

Round 1
Reviewer 1 Report
The work is of interest to the community and is well and technically written. However, I have minor comments to be addressed before it is considered for publication.
1) The language of the manuscript needs a minor improvement.
2) why this review did not include the biodegradability of the surfactants?
3) What about the toxicity of surfactants?
4) details about the different applications did not include in the review.
Author Response
We thank the reviewer for the time taken to proofread our manuscript and for his comments. We will answer the reviewer's questions point by point. Our answers are in bold.
1) The language of the manuscript needs a minor improvement. : The manuscript has been proofread by a native speaker in order to improve its form
2) why this review did not include the biodegradability of the surfactants?
We don't understand the reviewer's comment. One part in our manuscript (4.4-Biological activity, cytotoxicity, and biodegradability) includes the biodegradability of the surfactants. If the reviewer finds that too much data is missing on this, then we agree with him. This is the observation that we make in this review where we specify that. In our opinion, there is a lack of information on this subject in the literature.
3) What about the toxicity of surfactants? Same answer as previous comment
4) details about the different applications did not include in the review. We don't understand the rewierver's comment. There is a two-page section dedicated to this point: 4.6-Applications of CNSL-based surfactants. Given the size of the review, we have chosen not to expand on this subject any further.
Reviewer 2 Report
The review presented is devoted to the synthesis of various surfactants based on very important bio-raw materials.
- Page 19, in Scheme 33 symbol Ph2N should be Ph2NH.
- Page 22, Line 635 "ethoxylated" should be "propoxylated".
- Sometimes minimum surface tension symbolized with upsilon. ϒCMC (Upsilon) should be change γCMC (gamma).
- Page 32, in Line 943-945 written “This point is illustrated by anionic molecules A-6 and A-6’ (see Figure 10), which comprise a PEO spacer between the phenol and the carboxylate: CMC decreases when the number of PEO units of the spacer increases.” However, A and B differ in the saturation and unsaturation of the alkyl chain (Line 1033-1034).
Author Response
We thank the reviewer for the time taken to proofread our manuscript and for his comments. We will answer the reviewer's questions point by point. Our answers are in bold.
1) Page 19, in Scheme 33 symbol Ph2N should be Ph2NH : The H has been added to Scheme 33 as specified by the reviewer. It was a typo.
2) Page 22, Line 635 "ethoxylated" should be "propoxylated". We have modified the text as requested by the reviewer.
3) Sometimes minimum surface tension symbolized with upsilon. ϒCMC (Upsilon) should be change γCMC (gamma). It has been modified as specified by the reviewer.
4) Page 32, in Line 943-945 written “This point is illustrated by anionic molecules A-6 and A-6’ (see Figure 10), which comprise a PEO spacer between the phenol and the carboxylate: CMC decreases when the number of PEO units of the spacer increases.” However, A and B differ in the saturation and unsaturation of the alkyl chain (Line 1033-1034).
We understand the reviewer's comment. The objective in this sentence is not to compare molecule A-6 to molecule A-6' but to show two examples where this phenomenon is visible. It happens for each of the two distinct series of molecules. The A-6 series and the A-6' series. In order to remove any possible doubt, we have modified this sentence by specifying that it is a series of distinct molecules whose value n can vary.